# Diversify Question Generation with Retrieval-Augmented Style Transfer

**Qi Gou[1], Zehua Xia[1], Bowen Yu[2], Haiyang Yu[2], Fei Huang[2]**
**Yongbin Li[2]\* and Cam-Tu Nguyen[1]\***

[1]State Key Laboratory for Novel Software Technology, Nanjing University, China
[2]Alibaba Group
{qi.gou,zehuaxia}@smail.nju.edu.cn
{yubowen.ybw, yifei.yhy, f.huang}@alibaba-inc.com
shuide.lyb@alibaba-inc.com ncamtu@nju.edu.cn

## Abstract

Given a textual passage and an answer, humans are able to ask questions with various expressions, but this ability is still challenging for most question generation (QG) systems. Existing solutions mainly focus on the internal knowledge within the given passage or the semantic word space for diverse content planning. These methods, however, have not considered the potential of external knowledge for expression diversity. To bridge this gap, we propose RAST, a framework for Retrieval-Augmented Style Transfer, where the objective is to utilize the style of diverse templates for question generation. For training RAST, we develop a novel Reinforcement Learning (RL) based approach that maximizes a weighted combination of diversity reward and consistency reward. Here, the consistency reward is computed by a Question-Answering (QA) model, whereas the diversity reward measures how much the final output mimics the retrieved template. Experimental results show that our method outperforms previous diversity-driven baselines on diversity while being comparable in terms of consistency scores. Our code is available at https://github.com/gouqi666/RAST.

## 1 Introduction

Question Generation (QG) aims to generate questions from a given answer and a grounding paragraph. As a dual task of Question Answering (QA), QG can potentially be used for the automatic construction of QA datasets, thereby improving QA with little annotation effort (Shakeri et al., 2020; Alberti et al., 2019; Cui et al., 2021). Furthermore, QG can be utilized for educational purposes (Yao et al., 2022; Qu et al., 2021), dialog systems (Wu et al., 2022), and conversational recommendation systems (Montazeralghaem and Allan, 2022).

QG systems are typically known to suffer from two major issues, namely inconsistency and lack

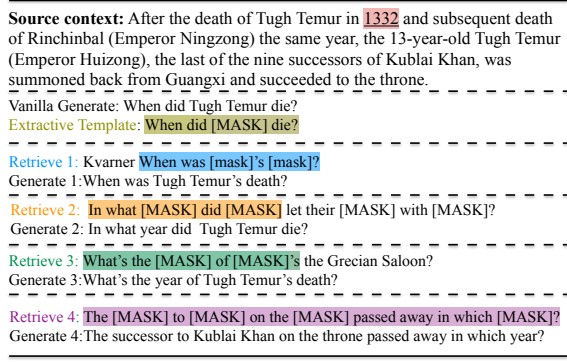

Source context: After the death of Tugh Temur in 1332 and subsequent death of Rinchinbal (Emperor Ningzong) the same year, the 13-year-old Tugh Temur (Emperor Huizong), the last of the nine successors of Kublai Khan, was summoned back from Guangxi and succeeded to the throne.

Vanilla Generate: When did Tugh Temur die?
Extractive Template: When did [MASK] die?

Retrieve 1: Kvarner When was [mask]'s [mask]?
Generate 1: When was Tugh Temur's death?

Retrieve 2: In what [MASK] did [MASK] let their [MASK] with [MASK]?
Generate 2: In what year did Tugh Temur die?

Retrieve 3: What's the [MASK] of [MASK]'s the Grecian Saloon?
Generate 3: What's the year of Tugh Temur's death?

Retrieve 4: The [MASK] to [MASK] on the [MASK] passed away in which [MASK]?
Generate 4: The successor to Kublai Khan on the throne passed away in which year?

Figure 1: Given a passage and an answer (in red, underlined text), a base generation model can only produce questions with a single type of expression. Diversified questions, however, can be produced by rewriting the base question using alternative (retrieved) templates. Note that the rewriting model should be robust to the noise existing in the retrieved templates.

of diversity. The former indicates that QG systems may yield context-irrelevant or answer-irrelevant questions (Zhang and Bansal, 2019; Zhao et al., 2018; Liu et al., 2019; Song et al., 2018a). The latter is because QG systems may fail to capture the one-to-many nature of QG tasks; that is, many questions can be asked given the same pair of context and answer. Existing solutions mainly exploit the internal knowledge within the context (Narayan et al., 2022a; Wang et al., 2020b), the language model (Fan et al., 2018a; Holtzman et al., 2019), or the semantic word space (Shen et al., 2019; Cho et al., 2019) for diverse content planning. Unfortunately, since these methods rely on obscure factors such as the black-box language model or the latent variable, they are not as controllable as exploiting external question templates (Figure 1).

In this paper, we aim to improve generation diversity by looking for expression variations in an external set of question templates. Figure 1 shows several questions that can be generated with a number of retrieved templates for a given source context.

---
\*Corresponding authors.

Although external information has been exploited for QG (Cao and Wang, 2021; Deschamps et al., 2021), prior methods depend on manually crafted set of type-dependent templates (Cao and Wang, 2021) or paraphrasing samples (Deschamps et al., 2021). In contrast, we neither require the annotation of question types and the relevant templates (unlike Cao and Wang (2021)) nor assume that question rewriting samples are accessible (unlike Deschamps et al. (2021)).

Our framework contains three main components: (1) a vanilla generator for initial template planning; (2) a style retriever, which filters related style templates given the initial one; and (3) a style-based generator, which robustly combines a style template and the internal context to generate the final question. Training such a model, however, is non-trivial due to two issues: 1) diversity should not come with the cost of consistency; 2) the lack of template-based question rewriting samples. We address these issues with Reinforcement Learning (RL), which directly maximizes a balanced combination of consistency and diversity rewards. Here, the consistency metric is computed by a Question-Answering (QA) model, whereas the diversity metric measures how much the final output mimics the retrieved template. Unlike the standard maximum likelihood approach, we do not need token-by-token supervised signals for training with RL, thus relaxing the need for question rewriting samples. Our approach is inspired by the retrieval-and-edit methods (Cai et al., 2019a,b), but focuses on the unexplored problem of balancing diversity and consistency by using RL.

All in all, our main contributions are three-fold:

1. We propose RAST, a framework for Retrieval-Augmented Style Transfer, which retrieves question style templates from an external set and utilizes them to generate questions with diverse expressions.

2. We propose a novel RL-based method to jointly train the retriever and the style-based generator in RAST. Our method is potentially adaptable for other retrieval-augmented tasks such as document-grounded dialog systems (Feng et al., 2020; Fu, 2022).

3. Experimental results on NewsQA (Trischler et al., 2017) and two splits of SQuAD datasets (Zhou et al., 2017; Du et al., 2017) show that RAST achieves superior performance on diversity whereas being comparable in terms of consistency.

## 2 Related Work

**Question Generation** Early attempts on QG are rule-based (Kunichika et al., 2004; Mostow and Wei, 2009), which are inflexible and labor-intensive. In addition, such methods are not able to generate questions from a larger context. Sequence-to-sequence-based methods (Du et al., 2017; Kumar et al., 2019) are able to overcome such issues, leading to better results. Recently, supervised fine-tuning pre-trained language models (PLM) have shown to achieve significant improvement (Dong et al., 2019; Qi et al., 2020). These systems, however, mostly focus on consistency, whereas diversity is also essential for downstream tasks such as QA. Prior attempts at diversity can be divided into two main categories, those that make use of internal knowledge such as content selection (Cho et al., 2019; Shen et al., 2019; Wang et al., 2020b) and improved decoding (Fan et al., 2018a; Narayan et al., 2022a), and those that exploit external patterns (Deschamps et al., 2021; Cao and Wang, 2021). Our work falls into the latter category but attempts to do so without samples for question rewriting.

**Retrieval-Augmented Generation** There has been a growing interest in integrating (external) knowledge from a retrieval model into a parametric language model for text generation. Wu et al. (2019) propose a retrieve-then-edit paradigm, where an editor is trained to modify a retrieval result to produce a more consistent response. Cai et al. (2019a,b) exploit skeletons to diversify text generation outputs, where a skeleton is obtained by masking query-specific information in the text. The retrieval-augmented generation approach has also been used for task-oriented dialogs (Feng et al., 2020, 2021; Shuster et al., 2021; Fu et al., 2022; Gou et al., 2023; Zhang et al., 2023). These studies, however, either exploit surface matching methods (e.g. tf.idf) (Song et al., 2018b; Cai et al., 2019a,b; Wu et al., 2019) or separately train the retrieval with relevant labels (Shuster et al., 2021; Feng et al., 2020, 2021; Fu et al., 2022). The retriever, therefore, might not be optimal for generation.

Several studies have jointly trained the retriever and the generation model (Lewis et al., 2020; Hossain et al., 2020; Glass et al., 2022) , but they have mostly focused on consistency, not diversity.

**Reinforcement Learning for Generation** Reinforcement learning (RL) has been used for text generation to mitigate the exposure bias issue associated with the standard Supervised Learning (SL) approach. Here, the exposure bias refers to the fact that generation during inference relies on predicted tokens instead of ground-truth tokens as in training. Furthermore, instead of optimizing proxy losses as in SL approach, RL directly optimizes the quality of interest via RL rewards, thus bridging the evaluation gap between training and testing. Researchers have proposed various RL rewards for QG, including answerability (for question generation) (Liu et al., 2020), BLEU-4 and Word Mover Distance (WMD) (Wang et al., 2020a; Chen et al., 2020), naturalness (Fan et al., 2018b), consistency using a Question Paraphrase Probability (QPP), and a Question Answering Probability (QAP) (Zhang and Bansal, 2019; Hosking and Riedel, 2019; Yuan et al., 2017). Previous methods have primarily focused on evaluating the consistency of generated questions, while we seek to evaluate both consistency and diversity. Here, the diversity is achieved by training a retrieval model for a retrieval-augmented generation.

Our work is closely related to RetGen (Zhang et al., 2022). This method, however, differs from ours in several ways: 1) only the retrieval model is optimized using RL in RetGen, whereas both the retrieval and the generation model are updated end-to-end via our RL framework; 2) it uses the likelihood of ground truth generation outputs as returns to update the retriever while we combine consistency and diversity rewards.

**Text Style Transfer** Our objective of question style transfer bears some resemblance to text style transfer studies (Li et al., 2018; Xu et al., 2018; Hu et al., 2022). The main difference is that we do not have predefined style labels, whereas most studies in the style transfer literature rely on given labels such as positive/negative or formal/informal.

**Paraphrase Generation** Paraphrasing involves transforming a natural language sentence into a new sentence with the same semantic meaning but a different syntactic or lexical surface form. Although diversity can be obtained by paraphrasing generated questions, our setting is different from (He et al., 2020; Goyal and Durrett, 2020; Hosking et al., 2022). Specifically, question paraphrase datasets, such as (Fader et al., 2013; Wang et al., 2017), do not associate context with each pair of (sentence, paraphrased sample). As such, paraphrasing in these datasets can only focus on different word choice or syntactic modification of an input question. In contrast, our consistency reward allows generating questions as long as the answer is the same with the input question given the context. In other words, our method also pays attention to different clues of the context for QG diversity.

## 3 Methodology

### 3.1 Overview

QG aims to generate question $y$ given a paragraph $c$ and answer $a$, which we combine to form the context $x = \{c, a\}$ for convenience. To indicate the position of the answer $a$ in $x$, we wrap it in a special tag <HL>. Previous works such as (Narayan et al., 2022a) model $p(y|x)$ for QG, i.e they rely on the internal knowledge of the context or the language model for diversity. Instead, we model our diverse QG as follows:

$$
\begin{aligned}
p(y|x, \mathbf{Z}) \\
&= \mathbb{E}_{z_0, z \in \mathbf{z}}[p(z_0|x) \times p(z|z_0)p(y|z, x)] \\
&= \text{vanilla QG} \times \text{RAST}
\end{aligned}
$$

where $\mathbf{Z}$ denotes the external corpus of question style templates, and $z_0$ indicates the initial question template that can be predicted based on the context $x$. The intuition is that we choose the style templates from the external knowledge ($z \in top - k$ from $\mathbf{Z}$) that are close but not the same as $z_0$, and utilize them to generate questions with diverse styles. During training, for a given context $x$, we extract $z_0$ from the ground truth question $y$ by masking context-sensitive information. During inference, as we do not know the ground truth question, we rely on a vanilla question generation $p(y|x)$ (vanilla QG) to generate the best $y_0$ from which we extract the initial template $z_0$. In other words, we approximate $p(z_0|x) = 1$ for $z_0$ being the ground truth $z_0$ during training and the greedy $z_0$ of the vanilla QG during inference.

The general architecture of our framework is demonstrated in Figure 2, which contains a vanilla QG and a Retrieval-Augmented Style Transfer model (RAST model). It is noteworthy that although we apply a base T5 model (Raffel et al., 2020), many generation methods can be applied for vanilla QG to improve content diversity (Narayan et al., 2022a; Wang et al., 2020b), and the diversity

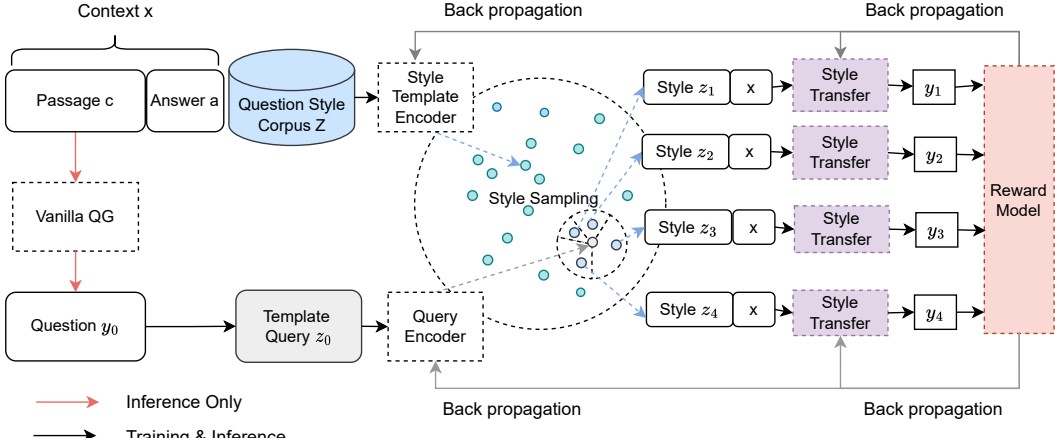

Figure 2: The architecture of our framework. During training, we use ground question $y_0$, while at evaluation, we use a vanilla QG model to generate a proxy question.

of RAST subsequently. The vanilla QG is trained using the standard maximum likelihood method, which we skip here for brevity. In the following, we detail our RAST model and how to train the model without samples for rewriting questions based on alternative styles ($z$).

## 3.2 Question Style Templates

To achieve style diversity in RAST, we use a set of question style templates which are constructed automatically through two steps - masking and duplication removal. Firstly, we leverage training data as our collected question corpus, allowing cross-sample reference for diverse question styles. The question templates are then obtained from the collected questions by masking context-sensitive information, making such patterns generalizable across contexts. Specifically, for each question, we keep stop and interrogative words, but replace entities (NER), noun phrases (NP), and context tokens with "[MASK]". Here, NER and NP are detected using Spacy[1]. Finally, near-duplicate templates are removed by measuring pairwise Jaccard similarities.

## 3.3 Retrieval-Augmented Style Transfer

**Style Retrieval Model**  We apply Dense Passage Retrieval (DPR) (Karpukhin et al., 2020) as the style retrieval model. Specifically, query and sample styles are encoded as the following:

$$
\begin{aligned}
q(z) &= BERT_1(z) & (1) \\
q(z_0) &= BERT_2(z_0) & (2) \\
p_\phi(z|z_0) &\propto \exp[q(z)^T q(z_0)] & (3)
\end{aligned}
$$

[1] https://spacy.io/usage/linguistic-features

where BERT-based encoders (Devlin et al., 2019) are used to convert question templates into dense embedding vectors for style retrieval. Sub-linear time search can be achieved with a Maximum Inner Product Search (MIPS) (Shrivastava and Li, 2014). Note that parameters of two encoders (2 BERT) constitute the parameter set $\phi$ of the style retrieval.

**Style Transfer Model**  We use T5 (Raffel et al., 2020) as our style transfer model $p_\theta(y|z, x)$, which generates questions auto-regressively based on a chosen style $z$ and the context $x$:

$$
p_\theta(y|x, z) = \prod_{i=1}^{T} p_\theta(y_t|x, z, y_{1:t-1}) \quad (4)
$$

where $T$ indicates the question length, and $\theta$ denotes T5 model parameters.

## 4 Two Stage Training

We train RAST using RL to avoid the exposure bias and the evaluation discrepancy between training and testing, which are often associated with supervised learning objectives (Chen et al., 2021). To accelerate the convergence of RL-based training, we first use supervised learning to initialize the style transfer model, resulting in a two-stage training procedure described in the following.

### 4.1 Supervised Learning

The style transfer model $p_\theta(y|x, z)$ can theoretically be initialized by the model trained on $\{(x, y_0, z_0)\}$, where $y_0$ is the ground truth question with the associated template $z_0$. Unfortunately, doing so results in an over-fitting model that is not

adaptable to training with noisy templates in the RL training phase. To overcome this issue, we actively corrupt $z_0$ to obtain a noisy template $\tilde{z}_0$ using several mechanisms, including (1) replacing [MASK] by a random entity; (2) adding some nouns; (3) deleting [MASK]; and (4) randomly choosing another template. Let $\hat{y}$ denote the predicted sequence given the input $x$ and the corrupted template $\tilde{z}_0$, the model is then trained with cross-entropy loss:

$$L_\theta^{CE} = -\sum_i y_i \log p(\hat{y}_i|x, \tilde{z}_0) \qquad (5)$$

where $y_i, \hat{y}_i$ denote the ground truth label and the predicted one at the time step $i$.

## 4.2 Reinforcement Learning

### 4.2.1 RL for Style Retrieval and Transfer

Our style retrieval and transfer problem are cast as a RL problem. Our model (RAST) introduced above can be viewed as an "agent" that interacts with an external "environment" of words and question templates. The parameters of the retrieval model ($\phi$) and the transfer model ($\theta$) define a combined policy that results in an "action" that is the selection of one style or the prediction of the next word. For simplicity, we assume that the style is chosen at the beginning of the sequence generation and kept unchanged throughout the generation process. Upon generating the end-of-sequence (EOS) token, the agent observes a "reward" $r$, which is detailed in Section 4.2.2. The goal of training is to minimize the negative expected reward:

$$L^{RL}(\theta, \phi) = -\mathbb{E}_{y^s \sim p_\theta, z^s \sim p_\phi}[r(y^s, z^s)] \quad (6)$$

where $y^s = (y_1^s, ..., y_T^s)$ and $y_t^s$ is the word sampled from the style transfer model $p_\theta$ at the time step $t$; $z^s$ is the template sampled from the style retrieval model $p_\phi$. Here, $y^s$ and $z^s$ are sampled according to the algorithm described in Section 4.2.3.

In order to compute the gradient $\nabla L^{RL}(\theta, \phi)$, we use REINFORCE method (Williams, 1992), which calculates a non-differential reward:

$$
\begin{aligned}
\nabla L^{RL} &= -\mathbb{E}_{y^s, z^s}[r(y^s, z^s)\nabla \log p_{\theta,\phi}] \\
&= -\mathbb{E}_{y^s, z^s}[r(y^s, z^s)\nabla \log p_\phi(z^s|z_0) - \\
&\quad \mathbb{E}_{y^s, z^s}[(r(y^s, z^s))\nabla \log p_\theta(y^s|x, z^s)] \\
&= \nabla L_\theta^{RL} + \nabla L_\phi^{RL} \qquad (7)
\end{aligned}
$$

where $p_{\theta,\phi}$ indicates $p_{\theta,\phi}(y^s, z^s|x, z_0)$, which can be decomposed into the product of the style transfer model $p_\theta(y^s|x, z^s)$ and the style retrieval model

$p_\phi(z^s|z_0)$. This subsequently decouples the gradients of the style transfer model $\nabla L_\theta^{RL}$ and the style retrieval model $\nabla L_\phi^{RL}$.

**RL with a Baseline and KL Divergence**    In order to reduce the variance of reinforcement learning for sequence generation, we modify the reward for the style transfer by referencing a baseline $b$ using the Self-critical sequence training (SCST) method (Rennie et al., 2017). Here, we use the reward of the greedy output of the style transfer model as the baseline, hence obtaining:

$$\nabla L_\theta^{RL} = -\mathbb{E}_{y^s, z^s}[(r(y^s, z^s) - b)\nabla \log p_\theta] \quad (8)$$

KL divergence is additionally used to avoid the updated policy ($p_\theta^*$) drifting too far away from the original one ($p_\theta$) (Liu et al., 2022; Schulman et al., 2017). The total gradient function for the style transfer model, therefore, is:

$$
\begin{aligned}
\nabla L_{\phi,\theta}^{RL} = \ &-\mathbb{E}_{y^s, z^s}[r(y^s, z^s)\nabla \log p_\phi] \\
&-\mathbb{E}_{y^s, z^s}[(r(y^s, z^s) - b)\nabla \log p_\theta] \\
&+\beta\nabla KL(p_\theta||p_\theta^*) \qquad (9)
\end{aligned}
$$

### 4.2.2 Reward Model

**Consistency Reward**    encourages the model to generate context-relevant and answer-relevant questions. Various strategies for consistency rewards can be used such as answerability (Liu et al., 2020), BLEU-4 and Word Mover Distance (WMD) (Wang et al., 2020a; Chen et al., 2020), naturalness (Fan et al., 2018b). In this paper, inspired by (Zhu and Hauff, 2021), we apply a Question Answer (QA) loss-based metric as our consistency reward. There are two reasons for QA-based metrics to be a good approximation for QG consistency: 1) QA is the dual task of QG; 2) the performance of QA systems, e.g., on SQuAD, has come close to human performance. Unlike (Zhu and Hauff, 2021), which uses an extractive QA model, we utilize a generative QA model based on T5 (Raffel et al., 2020). The reward is then measured as follows:

$$L_{qa} = -\frac{1}{T_a}\sum_{i=1}^{T_a} \log p(a_i|c, y^s, a_{<i}) \qquad (10)$$

$$r_{cons}(y^s, z^s) = \exp(-L_{qa}) \qquad (11)$$

where $T_a$ indicates the answer length, and $y^s$ is a sampled question from $p_\theta(y|z, x)$.

**Algorithm 1:** Diversity driven Sampling

**input :** The combination of paragraph and answer, $x = \{c, a\}$; the list of templates retrieved from $Z$ based on $z_0$, $S$; the generation sampling probability, $p$; and $k$.

**output :** $k$ sampled questions and styles

1   $clusters \leftarrow$ cluster $S$ into $k$ clusters based on Jaccard similarity;

2   $QZ^s \leftarrow$ empty set ;

3   **for** $i \leftarrow 1$ **to** $k$ **do**

4     **if** *training* **then**

5       $z^s \leftarrow$ randomly choose a style from $clusters[i]$;

6     **else**

7       $z^s \leftarrow$ select top style based on $p_\phi(z^s|z_0)$;

8     Sample $y^s$ from $p_\theta(y|x, z^s)$ using *nucleus sampling* with probability $p$;

9     Add $\{z^s, y^s\}$ to $QZ^s$;

10   **return** $QZ^s$

---

**Diversity Reward** promotes the generation of questions that are close to retrieved templates. For simplicity, we use Jaccard Similarity as our diversity reward as follows:

$$r_{divs}(y^s, z^s) = \frac{z^s \cap y^s}{z^s \cup y^s} \qquad (12)$$

**Total Reward** tries to trade off between consistency and diversity. It is obtained by combining the two rewards with a diverse coefficient $\lambda \in [0, 1]$ :

$$r(y^s, r^s) = r_{cons} + \lambda r_{divs} \qquad (13)$$

For the style transfer model, it is intuitive to see how this reward helps balance consistency and diversity. As for the style retriever, since the reward includes the consistency metric, we can assign higher scores to templates that can be used to generate various questions as long as the answer is $a$. By doing so, the style retrieval can go beyond surface matching and assign higher scores to templates of different styles.

### 4.2.3 Diversity-driven Sampling

One issue with training an RL model is that the model may degenerate to a locally optimal one, during which the retrieval puts all the probability mass on a small number of templates very close to $z_0$ according to surface matching, ignoring all

| Dataset | Train | Valid | Test |
|---------|-------|-------|------|
| SQuAD /1 | 86635 | 8965 | 8964 |
| SQuAD /2 | 70484 | 10570 | 11877 |
| NewsQA | 92549 | 5166 | 5126 |

Table 1: Statistic for datasets, where SQuAD /1 is the train/val/test split from (Zhou et al., 2017), and SQuAD /2 is another split from (Du et al., 2017).

the other templates. To overcome this, we propose a diversity-driven sampling procedure as in Algorithm 1. During training, we first cluster retrieved templates to group those close to each other according to surface matching (Jaccard similarity), then sample a template randomly from each cluster. By doing so, RL can have better exploration for various styles, thus avoiding the local optimal. During inference, however, we select the top template based on the retrieval scores from the well-trained retrieval model.

## 5 Experiments

### 5.1 Experiment Settings

**Datasets** We conduct experiments on two public datasets, SQuAD (Rajpurkar et al., 2016) and NewsQA (Trischler et al., 2017). As for SQuAD, since the test set is not accessible, we use the splits of (Zhou et al., 2017)[2] and (Du et al., 2017) instead. Table 1 provides the statistics of these datasets.

**Evaluation** Following (Wang et al., 2020b; Narayan et al., 2022b), we adopt several metrics to evaluate diversity and consistency: 1) *Top-1 BLEU* measures BLEU of the best generated output; 2) *Oracle BLEU* reflects the overall consistency by comparing the best hypothesis among top-N outputs with the target question. 3) *Pairwise BLEU* (or *Self BLEU*) measures the diversity by averaging sentence-level metrics of pairs within top N. A lower value of pairwise BLEU indicates a higher level of diversity. 4) *Overall BLEU* measures the overall performance, which can be calculated by Top-1 × Oracle ÷ Pairwise. Note that all the mentioned BLEU indicate BLEU-4.

### 5.2 Baselines

We compare our method with recent diverse-driven QG methods, which include those based on content

---
[2] https://res.qyzhou.me/redistribute.zip

| | Model | Top-1↑ | Oracle↑ | P-BLEU↓ | Overall↑ |
|---|---|---|---|---|---|
| SQuAD /1 | Mixture-Decoder (Shen et al., 2019) | 15.17 | 21.97 | 58.73 | 5.67 |
| | Mixture-Selector (Cho et al., 2019) | 15.67 | 22.45 | 58.82 | 5.88 |
| | CVAE (Wang et al., 2020b) | 15.34 | 21.15 | 54.18 | 5.99 |
| | Composition (Narayan et al., 2022a) ⋆ | 16.5 | **25.7** | 58.99 | 7.21† |
| | Nucleus-T5 (Holtzman et al., 2019) | 12.98 | 23.45† | 50.28 † | 6.05 |
| | **RAST(ours)** | **19.25** | 23.23 | **48.91** | **9.14** |
| uAD /2 | Composition (Narayan et al., 2022a) ⋆ | 15.94† | **24.90** | 60.05 | 6.61† |
| | Nucleus-T5 (Holtzman et al., 2019) | 13.31 | 24.42 † | **55.54** | 5.85 |
| | **RAST(ours)** | **19.36** | 22.59 | 56.42 † | **7.75** |
| NewsQA | Mixture-Decoder (Shen et al., 2019) | 10.02 | 17.04 † | 55.07 | 3.10 |
| | Mixture-Selector (Cho et al., 2019) | 10.90† | **17.51** | 52.61 | 3.63 |
| | CVAE (Wang et al., 2020b) | 9.90 | 15.48 | 41.37 | 3.70 † |
| | Nucleus (T5) (Holtzman et al., 2019) | 5.29 | 14.63 | 27.47 † | 2.82 |
| | **RAST(ours)** | **11.02** | 16.26 | **23.16** | **7.74** |

Table 2: Comparison of different techniques on question generation on NewsQA and two splits of SQuAD. Here, ⋆ denotes that the results are re-evaluated by us. The **best** and runner-up† are marked.

planning and those based on sampling.

**Content Planning-based Methods** Mixture Decoder (Shen et al., 2019) models a mixture of experts (MoE), where a latent variable drawn from MoE is used to control the generation and produce a diverse set of hypotheses. Mixture Selector (Cho et al., 2019) focuses on different parts of the context by modeling a binary variable for token selection. CVAE (Wang et al., 2020b) also selects tokens from the context, but uses a continuous latent variable instead of a binary variable like Mixture-Selector.

**Sampling-based Methods** Nucleus Sampling (Holtzman et al., 2019) samples tokens from a truncated distribution, where the unreliable tail of $1 - p$ probability mass is cropped. Composition Sampling (Narayan et al., 2022a) uses nucleus sampling to obtain diverse entity chains, then utilizes beam search to generate the most-likely output.

### 5.3 Implementation Details

We use the pre-trained DPR (Karpukhin et al., 2020) to initialize the retrieval encoders. Pre-trained T5-base [3] is used for vanilla QG and the style transfer model. During inference, the template of the vanilla QG is used as a query to retrieve $N - 1$ more templates. The obtained templates are then used to generate $N$ (N=5) questions for evaluation. We use SacreBLEU [4] to calculate BLEU.

[3] https://huggingface.co/t5-base
[4] https://github.com/mjpost/sacrebleu

More details can be found in Appendix A.

We conduct experiments with Nucleus-T5 by ourself using Transformers[5]. In addition, the results of Composition Sampling are reevaluated, whereas those of other baselines are from (Shen et al., 2019; Cho et al., 2019; Wang et al., 2020b).

### 5.4 Results and Analysis

Table 2 summarizes our experimental results, for which detailed analysis are given in the following.

**Among diverse-promoting baselines,** Nucleus-T5 promotes diversity with the cost of Top-1 BLEU being dropped significantly. CVAE and Composition are better at balancing between consistency and diversity, resulting in high overall scores. For example, in comparison with Nucleus-T5 on SQuAD /2, Composition is more consistent (better Top-1 and Oracle-BLEU), despite of being less diverse (lower Pairwise-BLEU). Our result is in line with (Narayan et al., 2022b).

**Compared to the previous methods,** RAST achieve the best diversity score (the lowest Pairwise-BLEU) on SQuAD/1 and NewsQA, and the second-best on SQuAD/2. Particularly, our method outperforms strong baselines (Composition Sampling and CVAE) by a large margin in terms of diversity, whereas being comparable on consistency scores. Specifically, on NewsQA and

[5] https://github.com/huggingface/transformers

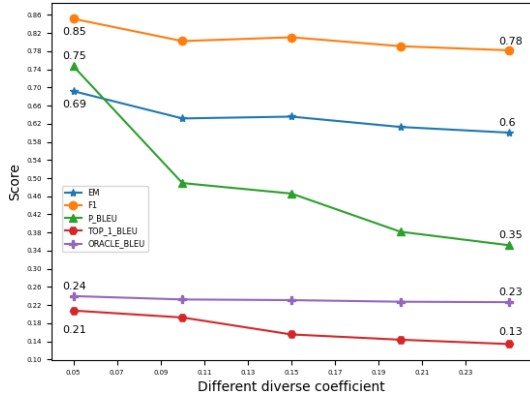

Figure 3: Results of changing the diverse coefficient $\lambda$ on SQuAD /1 (SQuAD v1.1 with split from (Zhou et al., 2017)). EM and F1 indicate QA-model based metrics, and P-BLEU is short for Pairwise-BLEU.

| Model | Top1↑ | Oracle↑ | P-B↓ | Over↑ |
|---|---|---|---|---|
| RAST | 19.25 | 23.23 | **48.91** | **9.14** |
| w/o e2e | 19.94 | 23.40 | 51.70 | 9.02 |
| w/o cluster | 15.58 | 23.04 | 61.06 | 5.88 |
| w/ question | 19.00 | **23.59** | 54.09 | 8.28 |

Table 3: Experimental results of different model variants on SQuAD /1 (Zhou et al., 2017). Here, P-B and Over are short for Pairwise-BLEU and Overall-BLEU respectively; EM and F1 indicate the QA-based metrics.

| Model | Consistency | Diversity | Total |
|---|---|---|---|
| RAST | **3.36** | **2.36** | **2.86** |
| Nucleus | 3.00 | 1.78 | 2.39 |

Table 4: Human evaluation result on SQuAD /1.

SQuAD/1, RAST is better than CVAE on both Top-1 and Oracle-BLEU. On SQuAD/1 and SQuAD/2, RAST is better than Composition on Top-1 whereas being comparable on Oracle-BLEU. Regarding the overall score, RAST obtains the superior results on three datasets, showing that its capability in balancing between diversity and consistency.

### 5.5 Ablation Study

We study the impact of different components of RAST on SQuAD/1 (Zhou et al., 2017), where the results are given in Figure 3 and Table 3.

**Diverse Coefficient** Figure 3 shows how diversity and consistency change when increasing $\lambda$. Besides Oracle-BLEU, we also use Exact Match (EM) and F1 (two QA metrics) to measure consistency (Sultan et al., 2020; Lyu et al., 2021). Here, the QA metrics are calculated by averaging EM and F1 scores of the answers, which are generated by the QA model for top-N evaluation questions.

As observable from Figure 3, increasing $\lambda$ leads to higher diversity (lower pairwise-BLEU), but lower consistency (lower Oracle and QA metrics). This is the result that we expect. The rate of increase in diversity, however, is much higher than the rate of decrease in the consistency metrics. Specifically, when $\lambda$ changes from 0.05 to 0.25, pairwise-BLEU drops 39.52 points (from 74.7 to 35.18), whereas F1 only drops 6.96 points (from 85.16 to 78.2), showing that our method is able to maintain consistency within a reasonable range while promoting diversity significantly.

**Freeze DPR** To study the impact of joint RL training on the retrieval and generation models, we

compare the performance of RAST and RAST (w/o e2e). As observable from Table 3, overall BLEU is improved with end2end training, showing that the retrieval model is better optimized for balancing between diversity and consistency.

**Diversity-driven Sampling** We measure the impact of the clustering step in diversity-driven sampling (Algorithm 1) by comparing RAST and RAST (w/o cluster) in Table 3. Here, during training, RAST (w/o cluster) samples templates based solely on the retrieval scores. It is observable that clustering allows us to better train RAST, and thus results in better performance across all metrics.

**Retrieval Query** The last row of Table 3 shows the performance of RAST when we use the best question $y_0$ of the vanilla QG (RAST w/ question) instead of the question template $z_0$ for querying external templates. As we can see, using masked questions (RAST) leads to higher diversity than the alternative. This is intuitive given the fact that masking context-sensitive information can make templates more generalizable across contexts.

### 5.6 Human Evaluation

We followed (Wang et al., 2020b) to evaluate the consistency and diversity of RAST and Nucleus sample[6] on 50 samples of SQuAD /1. Here, the consistency metric ranges from 0 to 5, measuring the proportion of the generated questions being answerable based on the given context (without

---

[6]This is because the source code of the other baselines is not publicly available

| Type | N-C | N-D | R-C | R-D |
|------|-----|-----|-----|-----|
| Fleiss' Kappa | 0.61 | 0.60 | 0.62 | 0.75 |

Table 5: Our inter-annotator agreement score. Here, N and R denote Nucleus and RAST, whereas C and D means consistency and diversity respectively.

---

**Source context:** J. A. Hobson identifies this justification on general grounds as: "It is desirable that the earth should be peopled, governed, and developed, as far as possible, by the races which can do this work best, i.e. by the races of highest social efficiency".

| | |
|---|---|
| **Retrieved Template:** | Which [MASK] to have [MASK] of [MASK]? |
| **RAST:** | Which race does Hobson believe to have the responsibility of human development? |
| **Retrieved Template:** | Who would be seen as having been [MASK] in the [MASK]? |
| **RAST:** | Who would be seen as having been best in the development of the earth? |
| **Retrieved Template:** | [MASK] states that [MASK] has the [MASK] to [MASK]? |
| **RAST:** | Hobson states which race has the ability to develop the earth to the best of its ability? |

Figure 4: The three RAST outputs with different question types. Here the given answer is highlighted with red color in the source context.

any hallucinations on the named entity or intent errors). On the other hand, the diversity metric calculates the number of distinct questions among the consistent ones, which means the diversity score ranges from 1 to the consistency score. Specifically, each sample has been checked by three annotators. The results in Table 4 indicate that RAST less suffers from hallucination, whereas being more diverse. We also provide our inter-annotator agreement score in Table 5, which indicate moderate to substantial agreement among our annotators.

## 5.7 Case Analysis

To better analyze the performance of RAST, we provide a case study in Figure 4. As shown in this case, RAST obtains its diversity by retrieving different templates. Notably, the third output replaces "that" in the template with "which," demonstrating that our model does not simply copy syntactic words from the template. Interesting, the diversity also results from selecting different clues from the context that are suitable for the retrieved templates, such as "Hobson," "race," "best," and "the development of the earth." Please refer to Appendix B for more cases.

## 6 Conclusion

This paper proposes RAST, a framework that exploits question templates from an external corpus to improve expression diversity in QG systems. Compared to previous methods, that exploit internal knowledge of language models for diversity, RAST provides a more flexible and interpretative way to control the generation outputs. To train RAST without question rewriting samples, we develop a novel RL based method, where we directly optimize the combination of the consistency and diversity rewards. In addition, we provide two stage training and diversity-driven sampling, which help better train our RL-based model. Experiment results show that RAST outperforms strong baselines in terms of diversity whereas being comparable on consistency scores. For future studies, we aim at further improvement by developing efficient training with a small number of paraphasing samples.

## Limitations

Our study currently suffers from several limitations: (1) QG evaluation is challenging due to one-to-many nature of the task. The best evaluation should be human evaluation. Unfortunately, this is not possible since we do not have access to source code of many previous studies. Although the outputs of Composition Sampling are available, they only come with the paired gold questions. Since the data was shuffled, we do not know the corresponding passages for human evaluation. As an alternative, we have tried to cover as many metrics as possibles, including all of the metrics used in previous baselines and QA-based metrics. (2) Training a RL-based method like RAST is typically more difficult and time consuming. This is because RL requires many rounds of sampling to converge. Our two-stage training is helpful, but there is still more room for improvement. (3) Our model is limited by the maximum context length like most of the Transformer-based methods.

## Ethics Statement

This paper uses opensource datasets to construct the external style corpus. One concern is that model can learn to mimic target properties in the training data that are not desirable. Another concern is that our work might involve the same biases and toxic behaviors in the pre-trained models.

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

| Parameters | SQuAD/1 | SQuAD/2 | NewsQA |
|---|---|---|---|
| g-lr | 1e-6 | 1e-6 | 1e-6 |
| d-lr | 1e-7 | 1e-7 | 1e-7 |
| max-len | 128 | 512 | 1250 |
| BS | 12 | 8 | 2 |
| T-num | 3 | 3 | 2 |
| Training hour | 48 | 70 | 120 |
| $\lambda$ | 0.5 | 0.5 | 0.4 |
| $\beta$ | 0.1 | 0.1 | 0.05 |

Table 6: Hyper-parameters of RAST at reinforcement learning stage, g-lr means generator learning rate, d-lr means DPR learning rate, BS means batch size, T-num means how many templates will be sent to style transfer model for every single training data, $\lambda$ is diverse coefficient and $\beta$ is coefficient of KL divergence.

# A Technical Details

## A.1 Implementation Details

Our model is implemented with Pytorch 1.8.1 and Transformers 4.23.1. For three datasets, we set max length of input as 128/512/1250 for SQuAD split1, SQuAD split2, and NewsQA respectively.

During inference, the template of the vanilla QG ($z_0$) is used as a query to achieve $N-1$ more templates, which are then combined with $z_0$ to generate $N$ questions for top-N evaluation (N=5). The $z_0$, however, is not actually be used for generating questions with the style transfer model, instead we replace it with an empty string. In other words, the input the style transfer model contain only context. This is done so that we do not take the advantage of the vanilla QG into account.

### A.1.1 Hyperparameters

During SL, we fine-tune the baselines for 5 epochs with learning rate of 5e-4 and 5 epochs. We set the sampling parameters with top-p of 0.9 and top-k of 30. Warmup-ratio and weight-decay are set as 0.1 for all three datasets. We set batch size as 64/32/6 for SQuAD/1, /2, and NewsQA, respectively.

For RL, we train RAST with 7 epochs and warmup-ratio of 0.2. The number of retrieval is set as 100 at training and 500 at evaluation. We choose 5 style templates for style transfer model at evaluation since we should calculate Oracle BLEU(K=5) with baselines for fair comparison. The final model is the one with the highest Oracle BLEU on development set. Please refer to Table 6 for more information.

# B Samples of Generation Results

| SQuAD v1.1, split1 |
|---|

**Input**: After the death of Tugh Temür in<HL> 1332 <HL> and subsequent death of Rinchinbal (Emperor Ningzong) the same year, the 13-year-old Toghun Temür (Emperor Huizong), the last of the nine successors of Kublai Khan, was summoned back from Guangxi and succeeded to the throne.

**GOLD:** When did Tugh Temur die?

**RAST:** In what year did Tugh Temür die?

When did Tugh Temür pass away?

When did Tugh Temür die?

When did Tugh Temür receive his last death?

Tugh Temür was killed when?

**Nucleus** In what year did Tugh Temür die?

When did Tugh Temür die?

When did Tugh Temür die?

In what year did Tugh Temür die?

In what year did Tugh Temür die?

**Input**: All chloroplasts in a plant are descended from<HL> undifferentiated proplastids <HL> found in the zygote, or fertilized egg.

**GOLD:** What do a plant's chloroplasts descend from?

**RAST:** What do all chloroplasts in a plant descended from?

Chloroplasts are typically made up of what type?

What do chloroplasts appear as in the zygote?

What type of organism in zygote usually recognized as descended from?

What type of organism were seen in a zygote?

**Nucleus:** What are all chloroplasts in a plant descended from?

What are all chloroplasts in a plant descended from?

What are all chloroplasts in a plant descended from?

What are all chloroplasts in a plant descended from?

What are all chloroplasts in a plant descended from?

**Input**: The merger was suspended, and a complaint was filed by the Department of Justice in July 1967, with ITT going to trial in October 1967; the merger was officially canceled after the trial's conclusion on<HL> January 1, 1968 <HL>

**GOLD:** When was the merger between ITT and ABC officially canceled?

**RAST:** When did the trial of ITT end?

On what date was the merger officially canceled?

The merger was not officially canceled until when?

When was the trial where the merger of ITT appeared?

The merger became the canceled when?

**Nucleus:** When was the merger officially canceled?

When was the trial for ITT's merger officially canceled?

When did the court's decision about the ITT merger come to an end?

When was the merger officially canceled?

When did ITT's merger with ITC officially end?

Figure 5: Examples of input context, ground truth question, and model predictions for SQuAD v1.1 dataset of split (Zhou et al., 2017). We also show the result generated by nucleus sampling of which P-BLEU is similar with RAST while Top-1 BLEU and Oracle BLEU are lower than RAST. Despite similar values of P-BLEU, the interrogative words and syntactic structures generated by nucleus sampling are homogenous, not as diverse and flexible as RAST.