# OpenReview forum: "Diversify Question Generation with Retrieval-Augmented Style Transfer"
_EMNLP/2023/Conference — EMNLP 2023 Main_

### Official Review · Reviewer_nwXM · 2023-07-31

**Soundness:** 4

**Excitement:**

3: Ambivalent: It has merits (e.g., it reports state-of-the-art results, the idea is nice), but there are key weaknesses (e.g., it describes incremental work), and it can significantly benefit from another round of revision. However, I won't object to accepting it if my co-reviewers champion it.

**Missing References:**

Learning Sparse Prototypes for Text Generation, He et al. (2020)

Machine Comprehension by Text-to-Text Neural Question Generation, Yuan et al. (2017)

Evaluating Rewards for Question Generation Models, Hosking & Riedel (2019)

Neural Syntactic Preordering for Controlled Paraphrase Generation, Goyal & Durrett (2020)

Hierarchical Sketch Induction for Paraphrase Generation, Hosking et al. (2022)

**Paper Topic And Main Contributions:**

The paper proposes improving the diversity of question generation systems by using retrieved templates as guidance for the rough syntactic structure of the generated question. The proposed retriever and “rewriting” model are trained end-to-end via RL.

**Questions For The Authors:**

A) Is the diversity reward on L370 invariant to word order? If so, wouldn’t this ignore syntactic diversity where the word order changes but the types of phrase used in the template stay the same?

B) How is a single vector extracted from the BERT encodings? I assume the use [CLS] token is used, but it is not explicitly mentioned in the paper.

C) What is the difference between Top-1 Bleu and Oracle BLEU? As far as I understand, both measure the maximum BLEU between all candidate generations and the reference question?

D) Are question templates retrieved based on their full surface form, or based on their template? If they are retrieved based on template, wouldn’t this return templates that are syntactically similar to the query, and therefore not diverse?

**Reasons To Accept:**

The paper correctly evaluates on the two different splits of SQuAD for QG.

The paper includes ablation studies, and a human evaluation (although this is limited to two systems)

**Reasons To Reject:**

There is no comparison to or discussion of paraphrasing approaches - the authors state that a strength of their approach is that no diverse references are needed, which is valid. But, there are a number of resources available for factual question paraphrasing (eg Paralex, QQP) so lack of supervision data is not an issue in factual QG. There is also insufficient engagement with prior work on retrieving and editing templates (see eg He et al. (2020)).

The authors claim in the abstract and limitations that their method is comparable in terms of consistency scores as measured by a QA model, but the QA score evaluation is missing from the results.

The reference-based evaluation metrics are poorly explained and are not standard. BLEU, Self-BLEU and iBLEU are commonly used in paraphrase generation and would be appropriate here.

Some evidence that the diversity introduced by the proposed method leads to improved downstream performance (eg when training a QA model on a new domain) would

The paper could be more clearly written - there are a number of minor typos, unclear paragraphs and missing details.

**Reproducibility:**

3: Could reproduce the results with some difficulty. The settings of parameters are underspecified or subjectively determined; the training/evaluation data are not widely available.

**Reviewer Confidence:**

3: Pretty sure, but there's a chance I missed something. Although I have a good feel for this area in general, I did not carefully check the paper's details, e.g., the math, experimental design, or novelty.

**Typos Grammar Style And Presentation Improvements:**

L202 - please explain how exactly the context and answer are combined.

L203-206 - this sentence does not make sense. Also, p(y|x)p(y|x) = p(y|x)^2, I’m not sure what this is supposed to show.

L208 - why does \textbf{Z} not appear on the right hand side of the equation? Please be more precise with the notation used here.

L222 - p(y|x)p(y|x) appears again.

L256 - please explain in a little more detail how similar templates are removed

It would be useful to include equation numbers.

Please tidy up Figure 3 - the x axis label and legend texts could be more precise

Paragraph 5.6 could be more clearly written

It is not clear whether template retrieval happens based on “full” questions or based on templates. Notation could be used more precisely to make this more clear.

---

> ### Author Rebuttal · Authors · 2023-08-29
>
> **Thank you for your careful review and thoughtful suggestions.** We provide our response to your questions and concerns as follows.
>
> Q1: There is no comparison to or discussion of paraphrasing approaches - the authors state that a strength of their approach is that no diverse references are needed, which is valid. But, there are a number of resources available for factual question paraphrasing (eg Paralex, QQP) so lack of supervision data is not an issue in factual QG. There is also insufficient engagement with prior work on retrieving and editing templates (see e.g. He et al. (2020)).
> > R1: Thank you for your comment. However, we think our method has its own value due to the following reasons:
> >   1) Paraphrasing and QG are different tasks. Specifically, there is **no context associated with each pair** of (sentence, paraphrasing) in Paralex and QQP. As such, paraphrasing can only focus on different word choice or syntactic modification. In contrast, our consistency reward allows generating questions as long as the answer is the same with the given one. In other words, we can find clues from context for variation of valid questions. Please refer to our response to Reviewer 2 for more information. We will update the related work to include the discussion on the relationship between paraphrasing and QG.
> >   2) Paraphrasing can help diversify questions in QG with **the cost of human effort on annotation.** Although the paraphrasing datasets such as Paralex or QQP are available, the cost for constructing both QA and QG and paraphrasing sets for new domains or new languages is not negligible. In addition, since the settings of two tasks are different as we previously mentioned, even in English, there is no alignment between (context, answer) in QA datasets and the paraphrasing pairs, hindering end-to-end supervised training.
> >
> >   Regarding He et al. (2020), this work is still paraphrasing. As such, it is not considered as a baseline even in previous QG studies (Want et al., EMNLP 2020, Jaemin et al., EMNLP 2019)
> [1] Diversify Question Generation with Continuous Content Selectors and Question Type Modeling, Wang et al., EMNLP 2020.
> [2] Mixture Content Selection for Diverse Sequence Generation, Jaemin et al., EMNLP 2019.
>
> Q2: The authors claim in the abstract and limitations that their method is comparable in terms of consistency scores as measured by a QA model, but the QA score evaluation is missing from the results.
> > R2: We are sorry for this confusion. In the abstract, we mean consistency and diversity rewards. In the limitation, we mean the evaluation of EM and F1 scores of QA model in Figure 3 in section 5.5. We will update our writing to make it clear.
>
> Q3: The reference-based evaluation metrics are poorly explained and are not standard. BLEU, Self-BLEU and iBLEU are commonly used in paraphrase generation and would be appropriate here.
> > R3: These metrics are commonly used for measuring diversity in QG ([1] Wang et al., EMNLP 2020) and ([2] Jaemin et al., EMNLP 2019). It is worth mentioning that **the original question required for calculating Self-BLEU and iBLEU measures is not available(only context and answer).** Therefore, it is not possible to calculate Self-BLEU and iBLEU in this context. We will update to write more detailed about these metrics.
> [1] Diversify Question Generation with Continuous Content Selectors and Question Type Modeling, Wang et al., EMNLP 2020.
> [2] Jaemin et al., Mixture Content Selection for Diverse Sequence Generation, EMNLP 2019.
>
> Q4: Some evidence that the diversity introduced by the proposed method leads to improved downstream performance (eg when training a QA model on a new domain) would
> > R4: Yes, the preliminary experiment shows that when we add new training samples from RAST, the performance of QA will increase.(F1:77.9->78.8). We will add more results in the final version.
>
> Q5: Is the diversity reward on L370 invariant to word order? If so, wouldn’t this ignore syntactic diversity where the word order changes but the types of phrase used in the template stay the same?
> > R5: Thank you for your comment, It is indeed invariant to word order. We  think that it is potential to exploit a diversity metric that encourages word order change, we will try it in the future work.
> >
> > &ensp;&ensp;&ensp;However, our current method can still generate diverse questions with word order change. This is because the retrieval is trained to retrieve different templates, which should maximize the consistentcy as well. In other words, the retrieval could get high reward for valid templates with different word order. By valid templates, we mean the corresponding questions can be responded with the same answer.
>
> Q6: How is a single vector extracted from the BERT encodings? I assume the use [CLS] token is used, but it is not explicitly mentioned in the paper.
> > R6: Yes, we use the first token [CLS] as the sentence embedding, We will update the paper in Line265 to make it clear.
>
> Q7: What is the difference between Top-1 Bleu and Oracle BLEU? As far as I understand, both measure the maximum BLEU between all candidate generations and the reference question?
> > R7: The two metrics are not the same. Top-1-bleu calculate the bleu between the top-1 output from model and gold question, whereas Oracle bleu is the maximum BLEU that can be achieved from k pairs of k candidates and the gold question. We adopt these metrics from (Jaemin et al., EMNLP 2019.)
> [1] Jaemin et al., Mixture Content Selection for Diverse Sequence Generation, EMNLP 2019
>
> Q8: Are question templates retrieved based on their full surface form, or based on their template? If they are retrieved based on template, wouldn’t this return templates that are syntactically similar to the query, and therefore not diverse?
> > R8: The question templates are retrieved using a dense retrieval model, which is trained end-to-end with the style transfer model for a diverse and consistent QG. As such, we do not retrieve based on the full surface form. For example, from figure 1, we can see that we can retrieve templates of “ in what [MASK] did [MASK] let their [MASK] with [MASK]?”, “The [MASK] to [MASK] on the [MASK] passed away in which {MASK}”given the template of “when did [MASK] died”.
>
> Q9：About Missing References:
> 1. Learning Sparse Prototypes for Text Generation, He et al. (2020)
> 2. Machine Comprehension by Text-to-Text Neural Question Generation, Yuan et al. (2017)
> 3. Evaluating Rewards for Question Generation Models, Hosking & Riedel (2019)
> 4. Neural Syntactic Preordering for Controlled Paraphrase Generation, Goyal & Durrett (2020)
> 5. Hierarchical Sketch Induction for Paraphrase Generation, Hosking et al. (2022)
>
> > R9: Thank you for your suggestions, ,we will add a paragraph on the difference of our task and the paraphrasing task as we previously described, and cite He et al. (2020), Goyal & Durrett (2020), Hosking et al. (2022). In addition, we will include the citation to the 2 papers. (Yuan et al., 2017 and Hosking, 2019)  in the part “Reinforcement Learning for QG” in Section II (related work). We have checked the papers and see that our discussion (line 177 to 178) on the difference of our work is still valid with these additional references.
>
> Q10: Typos Grammar Style And Presentation Improvements
> > - L202 - please explain how exactly the context and answer are combined.
>   **R:** we added a special token <HL> between the front and tail of answer in the context. Such as context.. <HL> answer <HL>...context.
>
> > -  L203-206 - this sentence does not make sense. Also, p(y|x)p(y|x) = p(y|x)^2, I’m not sure what this is supposed to show.  L222 - p(y|x)p(y|x) appears again.
> > **R:** We are sorry for this typo, it should be p(y|x). We will make sure to update it in the final version.
>
> > -  L208 - why does \textbf{Z} not appear on the right hand side of the equation? Please be more precise with the notation used here.
> >**R:**  Z is the whole corpus, since the space of Z is so large, we can’t take all data in consideration, so here we made an approximation where we just select top-k related styles through retrieval. We have explained that from in line 214-215. We will update the equation to reflect it.
>
> > - L256 - please explain in a little more detail how similar templates are removed
> >**R:** We calculated the Jaccard similarities between each pair templates, and remove one template for each pair, which has more than 70% similarity.
>
> > - Paragraph 5.6 could be more clearly written
> **R:** Thank you for your constructive comment, we will write it more clearly. In addition, we will add our discussion on the agreement of annotators. Specifically, each sample is checked by three people, and our Fleiss kappa are 0.6082, 0.6014, 0.6223, 0.7510 for Nucleus consistency, Nucleus diversity, RAST consistency and RAST diversity, respectively.
>
> > - It is not clear whether template retrieval happens based on “full” questions or based on templates. Notation could be used more precisely to make this more clear.
> **R:** in L211, we mentioned that the Z denotes the external corpus of question style templates, so we retrieve based on templates.
>
> > - It would be useful to include equation numbers,  Please tidy up Figure 3 - the x axis label and legend texts could be more precise
> **R:** Thank you, we will polish our papers according to your suggestion.

---

### Official Review · Reviewer_Tuqy · 2023-08-07

**Soundness:** 4

**Excitement:**

3: Ambivalent: It has merits (e.g., it reports state-of-the-art results, the idea is nice), but there are key weaknesses (e.g., it describes incremental work), and it can significantly benefit from another round of revision. However, I won't object to accepting it if my co-reviewers champion it.

**Paper Topic And Main Contributions:**

This paper aims to increase the diversity of automatically generated questions while maintaining their consistency in the context and answers. The proposed framework relies on question template retrieval and retrieval-augmented question generation, which is trained with the reinforcement learning method. Automatic and human evaluation results show that the proposed method outperforms existing work in terms of diversity while being comparable in terms of consistency.

**Questions For The Authors:**

1. Can you provide more case analysis showing how your proposed method is improving the *content/type* diversity of the generated questions besides their syntactic variability?

2. I'm missing details about the human evaluation process. How many annotators per sample are there in the human evaluation? Can you report the inter-annotator agreement?

**Reasons To Accept:**

1. The paper proposes a novel framework to improve the diversity of questions with carefully designed RL rewards and training stages.

2. The paper is overall clearly written and provides sufficient details to understand their framework.

**Reasons To Reject:**

1. Diversifying the generated question requires the system to find different angles for asking the question, rather than simply paraphrasing the question. However, template retrieval only increases the syntactical diversity but has no guarantee of making the question more grounded in multiple aspects of the context. This poses a limitation to the proposed method, which is not discussed in the paper.

2. Human evaluation is relatively weak. The proposed method is only compared to one simple baseline on one dataset. The results do not provide insight into why the proposed method is lagging behind other baselines in terms of consistency.

**Reproducibility:**

4: Could mostly reproduce the results, but there may be some variation because of sample variance or minor variations in their interpretation of the protocol or method.

**Reviewer Confidence:**

2: Willing to defend my evaluation, but it is fairly likely that I missed some details, didn't understand some central points, or can't be sure about the novelty of the work.

---

> ### Author Rebuttal · Authors · 2023-08-29
>
> We appreciate your constructive suggestions. We provide our response to your questions and concerns as follows.
>
> Q1: Diversifying the generated question requires the system to find different angles for asking the question, rather than simply paraphrasing the question. However, template retrieval only increases the syntactical diversity but has no guarantee of making the question more grounded in multiple aspects of the context. This poses a limitation to the proposed method, which is not discussed in the paper.
> > R1: RAST trains the bold text retrieval and the style transfer model in an end-to-end manner to optimize both the consistency reward (QA) and the diversity reward (Jaccard similarity). The consistency reward encourages both the retrieval and the style transfer models to generate faithful and valid questions that QA system can generate the same answer (the given answer). So, if the diversity reward encourages syntactic diversity, **the consistency score can help the system to ground on different aspects of the context as long as the answer is consistent.** Please refer to our case analysis below. And we will update our writing to make this clear.
>
> Q2: Human evaluation is relatively weak. The proposed method is only compared to one simple baseline on one dataset. The results do not provide insight into why the proposed method is lagging behind other baselines in terms of consistency.
> > R2: As we stated in our limitation, since we do not have access to the source code of previous baselines, we can only compare to the results on the automatic scores reported in the previous papers. Besides, human evaluation for diversity and consistency is time consuming. Specifically, we have to inspect outputs for each (context, question) pair for each method, metric that we measure (diversity and consistency). As such, we hope you see our efforts in providing automatic scores from multiple perspectives (Top-1 BLEU, Oracle BLEU, Pairwise BLEU, Overall BLEU, EM and F1) besides our human evaluation. It is noteworthy that our evaluation including human evaluation are in line with previous studies such as ([1] Wang et al., EMNLP 2020).
>      [1] Diversify Question Generation with Continuous Content Selectors and Question Type Modeling, Wang et al., EMNLP 2020.
>
> Q3: Can you provide more case analysis showing how your proposed method is improving the content/type diversity of the generated questions besides their syntactic variability?
> > R3: Thank you for your suggestions. We will update our paper to include case analysis as follows:
> > - Case 1.
> Given context : J. A. Hobson identifies this justification on general grounds as: "It is desirable that the earth should be peopled, governed, and developed, as far as possible, by the races which can do this work best, i.e. by <HL> the races of highest 'social efficiency" <HL>.
> >   1. Retrieved template: Which [MASK] to have [MASK] of [MASK] ?
>   RAST output: Which race does Hobson believe to have the responsibility of human development?
> >   2. Retrieved template: Who would be seen as having been [MASK] in the [MASK] ?
>   RAST output: Who would be seen as having been best in the development of the earth?
> >   3. Retrieved template: [MASK] states that [MASK] has the [MASK] to [MASK] ?
>   RAST output: Hobson states which race has the ability to develop the earth to the best of its ability?
> >
> >    In this sample, the three RAST outputs have different question types. They each select different clues from the context, such as "Hobson," "race," "best," and "the development of the earth." Importantly, the third output replaces "that" with "which," demonstrating that our model is not simply copying from the template.
>
> > - Case 2.
> Given context : "On 7 January 1943, at the age of <HL> 86 <HL> , Tesla died alone in room 3327 of the New Yorker Hotel."
> >   1.	Retrieved template:  How old was [MASK] until his [MASK] ?
> RAST output:  How old was Tesla's age until his death?
> >   2.	Retrieved template:  How old was [MASK] when he [MASK] in [MASK]?
> RAST output:  How old was Tesla when he died in New York?
> >   3.	Retrieved template: What was [MASK] at the time of [MASK]
> RAST output: What was Tesla's age at the time of his death?
> >
> >   In this sample, the three RAST outputs have different question types. They each capture different clues from the context, such as "in New York." As we can observe, the model is able to combine different parts of the template, such as "until" and "at the time of." The question types are also diverse.
>
> >- Case 3:
> Given context: "The merger was suspended, and a complaint was filed by the Department of Justice in July 1967, with ITT going to trial in October 1967; the merger was officially canceled after the trial's conclusion on <HL> January 1, 1968 <HL> ."
> >   1.	Retrieved template:  The [MASK] was unknown until what [MASK]?
> RAST output:  On what date was the merger officially canceled?
> >   2.	Retrieved template: The [MASK] was not [MASK] until when?
> RAST output:  The merger was not officially canceled until when?
> >   3.	Retrieved template: When was the [MASK] where the [MASK] of [MASK] appeared ?
> RAST output:  When was the trial where the merger of ITT appeared?
> >
> >   In this sample, the three RAST outputs have different ways to ask questions. They each capture different clues from the context, such as "the trial" or "the merger officially canceled."
> > **Overall, the outputs demonstrate the model's ability to pick up on relevant details from the context and generate questions accordingly.**
>
> Q4: How many annotators per sample are there in the human evaluation? Can you report the inter-annotator agreement?
> > R4: Each sample has been checked by three annotators. Our Fleiss kappa are 0.608256
> , 0.601435, 0.622347, 0.751049 for Nucleus consistency, Nucleus diversity, RAST consistency and RAST diversity, respectively. We will update the writing to include such information.

---

### Official Review · Reviewer_Yvj5 · 2023-08-07

**Soundness:** 4

**Excitement:**

4: Strong: This paper deepens the understanding of some phenomenon or lowers the barriers to an existing research direction.

**Paper Topic And Main Contributions:**

What is this paper about:
This paper proposes a new framework RAST to introduce more diversity in question generation by exploitaing question templates from an external corpus. RAST is trained using a RL method in which the reward function is a combination of the consistency and diversity. Experimental results show that RAST outperforms previous models.

Main Contributions:
1. New framework to improve diversity in question generation.
2. State-of-the-art results.
3. Paper is well written and easy to follow.

**Reasons To Accept:**

1. New framework to improve diversity in question generation.
2. State-of-the-art results.
3. Paper is well written and easy to follow.

**Reasons To Reject:**

-

**Reproducibility:**

4: Could mostly reproduce the results, but there may be some variation because of sample variance or minor variations in their interpretation of the protocol or method.

**Reviewer Confidence:**

1: Not my area, or paper was hard for me to understand. My evaluation is just an educated guess.

---

> ### Author Rebuttal · Authors · 2023-08-29
>
> Thanks for your comments. If you have any question about this paper, please contact us.

---

### Meta-Review · Area_Chair_nbMv · 2023-09-18

**Recommendation:** 4

**Metareview:**

This paper proposes an approach for improving the diversity of question generation by retrieving templates from an external corpus and using them as guidance about the approximate syntactic structure of the generated question. The proposed RAST system is trained through RL, where in which the reward function is a combination of the consistency and diversity.

Experimental results show that the proposed model outperforming baselines, in both automated and human evaluation, in terms of diversity while being comparable in terms of consistency.

---

### Decision · Program_Chairs · 2023-10-07

**Decision:**

Accept-Main

**Comment:**

This paper proposes an approach for improving the diversity of question generation by retrieving templates from an external corpus and using them as guidance about the approximate syntactic structure of the generated question. The proposed RAST system is trained through RL, where in which the reward function is a combination of the consistency and diversity.

Experimental results show that the proposed model outperforming baselines, in both automated and human evaluation, in terms of diversity while being comparable in terms of consistency.